# Discrete GPCR-triggered endocytic modes enable β-arrestins to flexibly regulate cell signaling

Benjamin Barsi-Rhyne[1,2], Aashish Manglik[3,4,5]*, Mark von Zastrow[1,2,5,6]*

[1]Tetrad Graduate Program, University of California, San Francisco, San Francisco, United States; [2]Department of Psychiatry and Behavioral Sciences, University of California, San Francisco, San Francisco, United States; [3]Department of Pharmaceutical Chemistry, University of California, San Francisco, San Francisco, United States; [4]Department of Anesthesia and Perioperative Care, University of California, San Francisco, San Francisco, United States; [5]Quantitative Biology Institute, University of California, San Francisco, San Francisco, United States; [6]Department of Cellular and Molecular Pharmacology, University of California, San Francisco, San Francisco, United States

*For correspondence:
Aashish.Manglik@ucsf.edu (AM);
mark.vonzastrow@ucsf.edu (MZ)

Competing interest: The authors declare that no competing interests exist.

**Abstract** β-Arrestins are master regulators of cellular signaling that operate by desensitizing ligand-activated G-protein-coupled receptors (GPCRs) at the plasma membrane and promoting their subsequent endocytosis. The endocytic activity of β-arrestins is ligand dependent, triggered by GPCR binding, and increasingly recognized to have a multitude of downstream signaling and trafficking consequences that are specifically programmed by the bound GPCR. However, only one biochemical 'mode' for GPCR-mediated triggering of the endocytic activity is presently known – displacement of the β-arrestin C-terminus (CT) to expose clathrin-coated pit-binding determinants that are masked in the inactive state. Here, we revise this view by uncovering a second mode of GPCR-triggered endocytic activity that is independent of the β-arrestin CT and, instead, requires the cytosolic base of the β-arrestin C-lobe (CLB). We further show each of the discrete endocytic modes is triggered in a receptor-specific manner, with GPCRs that bind β-arrestin transiently ('class A') primarily triggering the CLB-dependent mode and GPCRs that bind more stably ('class B') triggering both the CT and CLB-dependent modes in combination. Moreover, we show that different modes have opposing effects on the net signaling output of receptors – with the CLB-dependent mode promoting rapid signal desensitization and the CT-dependent mode enabling prolonged signaling. Together, these results fundamentally revise understanding of how β-arrestins operate as efficient endocytic adaptors while facilitating diversity and flexibility in the control of cell signaling.

## Editor's evaluation

In the canonical model of G protein-coupled receptor desensitization and endocytosis, the unmasking of the β-arrestin C-terminus plays a crucial role. Here, Barsi-Rhyne and colleagues revise and extend this fundamental paradigm by showing that there is a second structural region, the β-arrestin C lobe, that governs GPCR endocytic behavior. Importantly, class A GPCRs that transiently interact with arrestins prefer this new mode while class B GPCRs, which form more stable interactions with arrestins, employ both the canonical and the newly identified endocytic mechanism. Intriguingly, the canonical mode enables prolonged signaling while the new mode promotes rapid desensitization.

## Introduction

β-Arrestins were named for their ability to desensitize signaling by binding to ligand-activated β-adrenergic receptors and physically blocking heterotrimeric G protein coupling (*Lohse et al., 1990*). This function is similar to the 'arrest' of signaling mediated by binding of visual arrestin to the light-activated G-protein-coupled receptor (GPCR) rhodopsin (*Kühn and Wilden, 1987*), and it has been subsequently generalized to many GPCRs (*Benovic, 2021*). β-Arrestins, unlike visual arrestin, have the additional ability to act as endocytic adaptor proteins by associating with clathrin-coated pits (CCPs) after binding to a GPCR (*Ferguson et al., 1996*; *Goodman et al., 1996*). This association promotes clustering of GPCR/β-arrestin complexes on the plasma membrane leading to the subsequent internalization of complexes via clathrin-mediated endocytosis. The prevailing current view is that all GPCRs trigger β-arrestins' endocytic activity in the same way, by displacing the β-arrestin C-terminus (CT) to unmask endocytic determinants contained therein (*Figure 1—figure supplement 1a*; *Milano et al., 2002*; *Nobles et al., 2007*; *Shukla et al., 2013*; *Xiao et al., 2004*).

Compelling experimental support for this mode of regulated endocytosis has accumulated over the past 25 years, including the identification of specific clathrin (CHC) and AP2 (AP2β)-binding elements present in the CT which promote the clustering of GPCR/β-arrestin complexes in CCPs (*Burtey et al., 2007*; *Edeling et al., 2006*; *Ferguson et al., 1996*; *Goodman et al., 1996*; *Kang et al., 2009*; *Kim and Benovic, 2002*; *Krupnick et al., 1997*; *Laporte et al., 2000*; *Orsini and Benovic, 1998*; *Schmid et al., 2006*). However, for most of this time, β-arrestins were thought to regulate signaling only from the cell surface. It is now clear that β-arrestins are more flexible regulators, and that they have the capacity to promote as well as attenuate signaling from the plasma membrane and endomembranes (*Calebiro et al., 2009*; *DeFea et al., 2000*; *Feinstein et al., 2013*; *Ferrandon et al., 2009*; *Irannejad et al., 2013*; *McDonald et al., 2000*; *Terrillon and Bouvier, 2004*). Furthermore, it is now also widely recognized that β-arrestins can have different effects on downstream signaling depending on GPCR-specific differences in the composition, stability, and/or conformation of the complex that they form with GPCRs (*Asher et al., 2022*; *Latorraca et al., 2020*; *Lee et al., 2016*; *Mayer et al., 2019*; *Nobles et al., 2011*; *Nuber et al., 2016*; *Yang et al., 2018*; *Yang et al., 2015*). In light of the high level of diversity and flexibility that is presently recognized to exist at the GPCR/β-arrestin interface, we wondered if release of the β-arrestin CT is sufficient to explain the diversity of downstream effects conferred by GPCR transit through CCPs.

Here, we show that release of the β-arrestin CT is only one mode by which GPCR binding can trigger the endocytic activity of β-arrestins. We resolve a second mode that is clearly distinct from the canonical CT-dependent mode because it does not require the β-arrestin CT whatsoever. We further show that GPCRs exhibit selectivity in triggering one or the other endocytic mode, and that each mode is differentially coupled to the GPCR/ β-arrestin interface to produce opposing effects on the net signaling output of receptors.

## Results

### The β-arrestin CT is dispensable for β2-adrenergic receptor internalization

The ability of the β-arrestin CT to associate with CCP components and influence GPCR endocytosis has been clearly established (*Burtey et al., 2007*; *Edeling et al., 2006*; *Ferguson et al., 1996*; *Goodman et al., 1996*; *Kang et al., 2009*; *Kim and Benovic, 2002*; *Krupnick et al., 1997*; *Laporte et al., 2000*; *Orsini and Benovic, 1998*; *Schmid et al., 2006*). However, to our knowledge, whether the β-arrestin CT is necessary for GPCR endocytosis has not been explicitly tested. To assess this, we pursued a genetic rescue strategy using CRISPR-engineered cells lacking both β-arrestins (βarr1/2 DKO) (*O'Hayre et al., 2017*) and asked if β-arrestin-2 (βarr2) is sufficient to rescue regulated endocytosis of the β2-adrenergic receptor (β2AR), a prototypic β-arrestin-dependent cargo. We verified that our experiments were carried out in an expression regime in which we confirmed that receptor internalization is phosphorylation dependent (Extended Data, *Figure 1g*), and began by examining agonist-induced clustering of receptors into CCPs on the plasma membrane as an essential intermediate step in the endocytic mechanism. We verified by total internal reflection fluorescence (TIRF) microscopy that β2ARs localize diffusely on the plasma membrane in βarr1/2 DKO cells, regardless of whether cells were exposed to the adrenergic agonist isoproterenol (Iso, *Figure 1a*). However, after

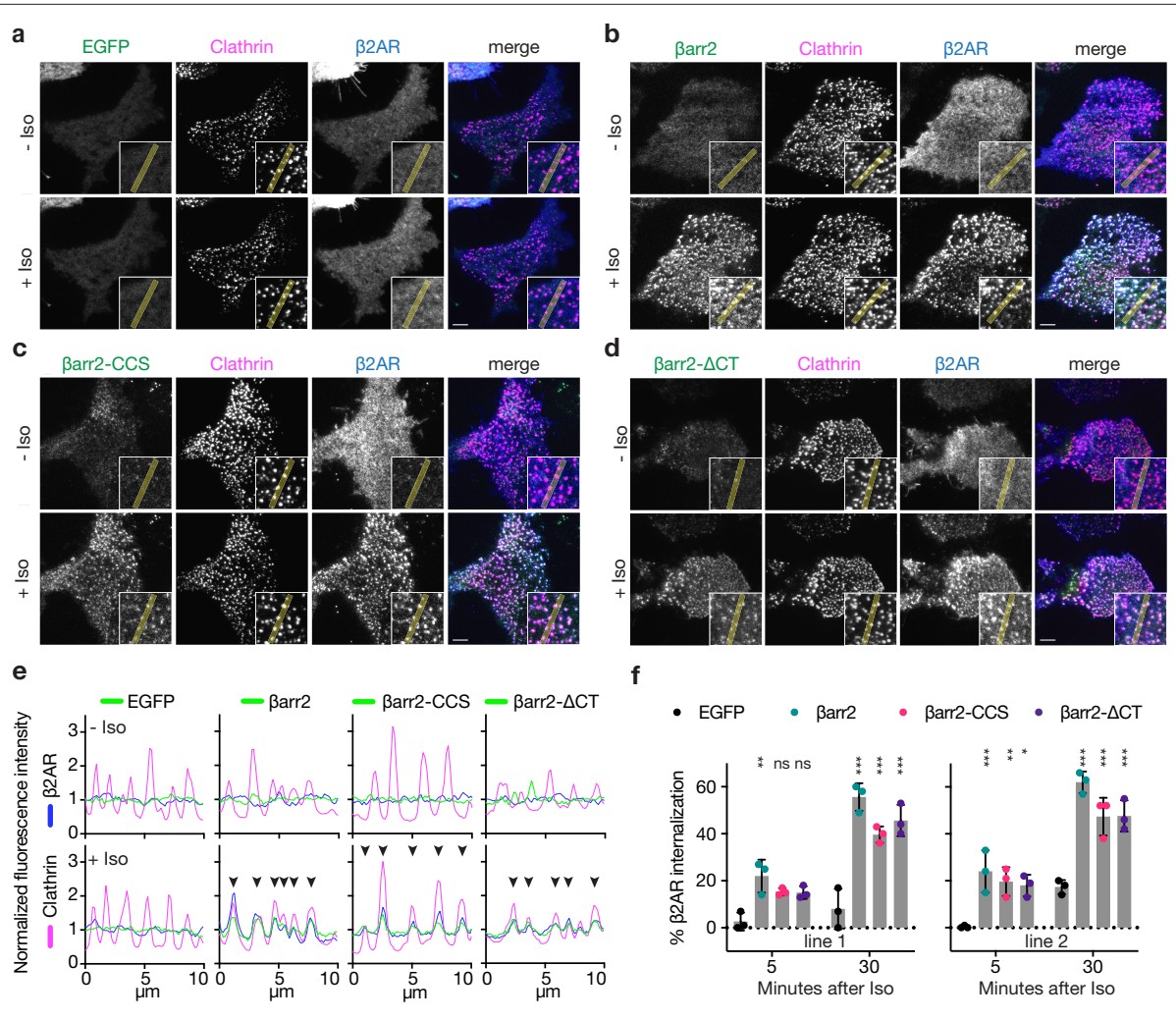

**Figure 1.** Known endocytic motifs in βarr2 are dispensable for β2-adrenergic receptor (β2AR) clustering and endocytosis. Representative live-cell total internal reflection fluorescence (TIRF) microscopy images of βarr1/2 double knockout HEK293 cells coexpressing clathrin-light-chain-DsRed (magenta) and FLAG-tagged β2AR (blue) with either EGFP (**a**), βarr2-EGFP (**b**), β arr2-CCS-EGFP (**c**), or βarr2-ΔCT-EGFP (**d**) (all in green) and pre- and post-stimulation with 10 μM isoproterenol (Iso). Scale bars are 5 μm. (**e**) Representative fluorescence intensity profiles from line scans shown in insets from a to d. Chevrons indicate colocalization. (**f**) Percent internalization of FLAG-tagged β2AR coexpressed with either EGFP (black), wild-type βarr2-EGFP (green), βarr2-CCS-EGFP (pink), or βarr2-ΔCT-EGFP in two clonal lines of βarr1/2 DKO HEK293 cells at 5- and 30-min post-stimulation with 10 μM isoproterenol (Iso). Data shown as mean ± standard deviation (SD) for $n$ = 3 independent experiments. Significance was determined by two-way analysis of variance (ANOVA) (df = 3, $F$ = 24.48) with Tukey's multiple comparisons test against the negative control (EGFP) for each time point (ns $p \geq 0.05$, *$p < 0.05$, **$p < 0.01$, ***$p < 0.001$). Each dot is an average of three technical replicates. All data shown are from three independent experiments.

The online version of this article includes the following source data and figure supplement(s) for figure 1:

**Source data 1.** Representative gating for all flow cytometry-based internalization assays.

**Figure supplement 1.** βarr2 and βarr1 C-terminus (CT) are dispensable for G-protein-coupled receptor (GPCR) internalization and β2-adrenergic receptor (β2AR) phospho-sites are required for efficient internalization.

**Figure supplement 1—source data 1.** Source data for the quantifications graphed in *Figure 1—figure supplement 1*.

**Figure supplement 2.** Representative unprocessed images of Coomassie stained sodium dodecyl sulfate/polyacrylamide gel electrophoresis (SDS/PAGE) gel and western blots.

expression of recombinant βarr2 (βarr2-EGFP), Iso-induced β2ARs to rapidly transition from a diffuse distribution to co-clustering together with βarr2 in CCPs (*Figure 1b*).

Surprisingly, genetic rescue of βarr2-driven clustering was observed even after clathrin and AP2-binding elements in the β-arrestin CT were disrupted using previously defined point mutations (βarr2-CCS construct) (*Eichel et al., 2018*). Confirming this, βarr2 rescued clustering and internalization

after both elements were fully removed by truncation (βarr2-ΔCT, truncated after N372) (*Figure 1c–e*). The latter result was verified in two independent βarr1/2 DKO cell lines (*Figure 1f*) and it was not limited to the β2AR (*Figure 1—figure supplement 1b–f*). Moreover, CT-independent trafficking was not unique to β-arrestin-2, as a β-arrestin-1 mutant lacking the CT (βarr1-ΔCT, truncated after N375) also promoted Iso-induced β2AR internalization (*Figure 1—figure supplement 1h*). In all cases, endocytosis is clathrin dependent (*Figure 1—figure supplement 1i*) and we verified that that removing the CT abrogated the previously identified interactions (*Figure 1—figure supplement 1j*). Taken together, these observations indicate that the β-arrestin CT is indeed not required for β-arrestin to drive clustering of GPCR/β-arrestin complexes on the cell surface or promote subsequent endocytosis of complexes. As both clustering and endocytosis mediated by βarr2-ΔCT remained agonist dependent, these results suggest that β-arrestins contain additional endocytic determinant(s) capable of mediating GPCR-triggered endocytic activity in the absence of the β-arrestin CT.

## A discrete endocytic determinant in the β-arrestin C-lobe

To investigate the nature of this additional endocytic activity, we took advantage of the fact that visual arrestin (v-arr) naturally fails to support GPCR endocytosis (*Moaven et al., 2013*) and asked if we could generate a v-arr/βarr2 chimera that contains the CT of v-arr but retains the ability to promote GPCR endocytosis due to the addition of other βarr2-derived sequence(s) (*Figure 2—figure supplement 1*). We tested this using the same genetic rescue strategy. To focus specifically on effects downstream of GPCR binding, we measured internalization using a chimeric mutant receptor (β2V2R chimera) that has higher affinity for binding arrestins than the β2AR (*Oakley et al., 2000*; *Oakley et al., 2001*). We were indeed able to generate such a chimeric mutant arrestin construct, here called ChiA, which rescued agonist-induced internalization of receptors nearly as strongly as wild-type βarr2 despite its entire CT being derived from v-arr (*Figure 2a*).

To identify sequence(s) responsible for the endocytic activity of this chimeric arrestin protein, we systematically replaced small sections of βarr2-derived sequence in ChiA to the corresponding sequence in v-arr (ChiA.1–14). Of 14 chimeras tested, we found three – ChiA.9, 10, and 11 – that failed to rescue internalization (*Figure 2a*, sequence details are in *Figure 2—figure supplement 1*). TIRF imaging indicated that all these internalization-defective chimeras were recruited to the plasma membrane in response to Iso addition, and to a similar degree as wild-type βarr2. However, they localized diffusely and failed to cluster receptors in CCPs. This was evident visually (*Figure 2b, c*) and quantified by fluorescence intensity measurement (*Figure 2d, e*). These data suggest that each of the mutations specifically interferes with the clustering and endocytic function of β-arrestin-2 by preventing accumulation in CCPs, but without affecting receptor-triggered recruitment to the plasma membrane. The endocytosis-blocking mutations mapped to a contiguous region of β-arrestin-2, located at the cytoplasmic face of the C-lobe and opposite the receptor-binding interface, which we called the β-arrestin C-lobe base (CLB, *Figure 2f*).

## The C-lobe determinant is essential for β2AR internalization

We next investigated the contributions of mutating the CLB or CT on the natural endocytic activity of βarr2. We focused on the central β-strand in the CLB of βarr2 and introduced mutations corresponding to residues present in v-arr (D205S, L208I, L215I, N216P, N218T, and H220A), while leaving the βarr2 CT unchanged. The resulting mutant construct, βarr2-CLB, was strongly recruited to the plasma membrane after Iso-induced activation and clustered together with β2AR in CCPs (*Figure 3b*). When the CLB was mutated in combination with the CT (βarr2-CLB,ΔCT mutant), agonist-induced recruitment to the plasma membrane was retained but βarr2-CLB,ΔCT was recruited diffusely and failed to promote surface clustering of β2ARs (*Figure 3d*). More detailed inspection of TIRF microscopy images revealed that mutating either the CLB or CT produced a partial reduction in clustering relative to wild-type βarr2, whereas clustering was abolished in the double mutant (*Figure 3e*). We verified this specific effect on clustering using a previously established quantitative metric (*Figure 3f, g*; *Eichel et al., 2018*), despite all of the constructs being similarly recruited to the plasma membrane (*Figure 3h*). We also assessed the ability of scFv30, a protein that recognizes a three-dimensional epitope in active βarr, to bind our mutants using an engineered split luciferase protein complementation system (NanoBiT, *Dixon et al., 2016*; *Shukla et al., 2013*; *Shukla et al., 2014*; *Xiao et al., 2004*). All of the βarr2 constructs robustly bound upon addition of an activating phosphopeptide (V2Rpp),

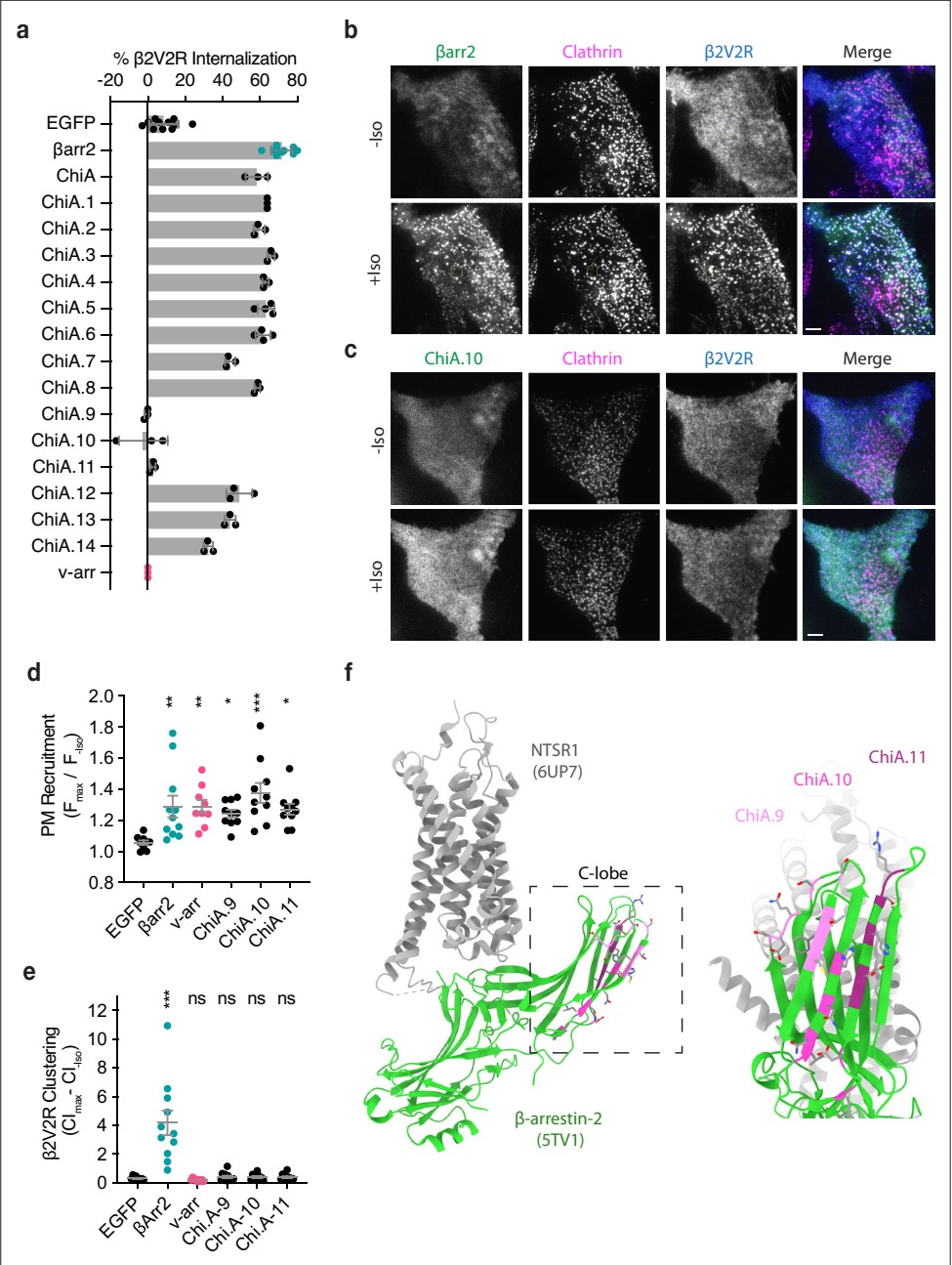

**Figure 2.** Identification of the βarr2 C-lobe base (CLB). (**a**) Percent internalization of β2V2R after 30 min of 10 μM isoproterenol stimulation in βarr1/2 DKO HEK293s coexpressing the indicated construct ($n \geq 3$ independent experiments, line is mean, error bars are standard deviation, each dot is an average of three technical replicates). Representative total internal reflection fluorescence (TIRF) microscopy images of cells expressing β2-adrenergic receptor (β2AR) (blue) and clathrin-light-chain-dsRed (magenta) with either wild-type βarr2-EGFP (**b**) or an example of one of the three internalization-defective chimeras, ChiA.10-EGFP (**c**) pre- and post-stimulation with 10 μM isoproterenol (Iso). Scale bars are 5 μm. (**d**) Plasma membrane recruitment of the indicated EGFP-tagged proteins (see Methods) in response to stimulation with 10 μM isoproterenol. (**e**) Maximum clustering index (CI, see Methods) of plasma membrane β2V2R after treatment with 10 μM Iso. For (**d, e**), each dot represents an individual cell. Data are shown as mean ± standard error of the mean (SEM) ($n \geq 9$ cells). Significance was determined by ordinary one-way analysis of variance (ANOVA) (df = 5 for both, $F$ = 22.21 and 4.531, respectively) with Dunnett's multiple comparison test against negative control (EGFP) (ns $p \geq 0.05$, *$p < 0.05$, **$p < 0.01$, ***$p < 0.001$). (**f**) Location of mutations unique to ChiA.9–11 (shades of pink and purple) in an active state structure of β-arrestin-2 (5TV1, green) (***Chen et al., 2017***) fit to the NTSR1/βarr1 structure (6UP7, gray) (***Huang et al., 2020***) (βarr1 not shown) and the

*Figure 2 continued on next page*

*Figure 2 continued*

same model rotated and zoomed to the cytoplasmic face of the C-lobe. All data shown are from at least three independent experiments.

The online version of this article includes the following source data and figure supplement(s) for figure 2:

**Source data 1.** Source data for quantifications graphed in *Figure 2*.

**Figure supplement 1.** Diagram of chimeras and sequence alignments of arrestins.

suggesting that the mutants are properly folded (*Figure 3—figure supplement 1a*).Together, these results indicate that the β-arrestin CT and CLB promote β2AR clustering in an additive manner. As βarr2-CLB,ΔCT failed to cluster at all, the results also indicate that the CT and CLB, together, fully account for the GPCR-triggered clustering activity of βarr2.

Despite the ability of the CT and CLB to promote β2AR clustering to a similar degree (*Figure 3g*), their effects on the subsequent internalization of receptors differed considerably. Mutating the CT only slightly reduced the ability of βarr2 to drive internalization of β2ARs, whereas mutating the CLB caused a pronounced reduction – suppressing the measured internalization of receptors to a level similar to that of the EGFP negative control (*Figure 3i*). We also found that our double mutant, βarr2-CLB,ΔCT, acts as a strong dominant negative when expressed at high levels in parental HEK293 cells containing endogenous βarr (*Figure 3—figure supplement 1b*). The corresponding mutations in the CLB of βarr1 (βarr1-CLB; D204S, S215P, N217T, H219A; *Figure 3—figure supplement 1c*) also failed to support β2AR internalization when expressed in DKO cells, indicating that this determinant functions similarly in both β-arrestins. βarr1 contains an additional endocytic determinant at an edge of the C-lobe (334-LLGDLASS-341) distinct from the CLB (*Kang et al., 2009*; *Staus et al., 2020*), but deleting this sequence did not significantly alter β2V2R internalization (*Figure 3—figure supplement 1d*). Remarkably, we found that changing a single solvent exposed residue in the CLB to the corresponding residue in visual arrestin, N218T, failed to support β2AR internalization when expressed DKO cells (*Figure 3—figure supplement 1e, f*). This represents a minor change, suggesting that the CLB may be a highly selective protein interaction surface. Taken together (*Figure 3j*), these data suggest that both the CLB and CT contribute to the total endocytic activity of β-arrestin, and each promotes agonist-dependent clustering of receptors on the cell surface. This indicates that, while both the CLB and CT are conserved in β-arrestins, the CLB is the primary determinant required for the efficient internalization of β2ARs.

## GPCRs differ in the degree to which they utilize each endocytic determinant

To assess the role of each determinant more broadly, we examined the effect of individually mutating the CT (βarr2-ΔCT) or CLB (βarr2-CLB) on agonist-induced internalization across several β-arrestin-dependent GPCRs. In an effort to include receptors differing in engagement at the GPCR/β-arrestin interface, we included three naturally occurring 'class A' GPCRs (β2AR, μ-opioid receptor or μOR, and κ-opioid receptor or κOR) that associate with β-arrestins relatively weakly and three 'class B' GPCRs (V2 vasopressin receptor or V2R, M2 muscarinic receptor or M2R, and neurotensin receptor type 1 or NTSR1) that bind β-arrestins more strongly (*Oakley et al., 2000*; *Oakley et al., 2001*). In addition, we included a chimeric GPCR containing the β2AR-derived transmembrane core and V2R-derived cytoplasmic tail (β2V2R) that is widely used as an experimental model of a class B GPCR.

Removing the β-arrestin-2 CT had little effect on agonist-induced internalization of the class A GPCRs, but significantly inhibited internalization of the naturally occurring class B GPCRs. By contrast, mutating the CLB inhibited internalization of all receptors tested, although the magnitude of this effect varied depending on the receptor (*Figure 4a*). Together, these results indicate that both the CT and CLB are broadly utilized endocytic determinants, but that individual GPCRs vary significantly in the degree to which their endocytosis depends on each.

Despite GPCR-specific differences in reliance on the CT relative to CLB, mutating both the CT and CLB in combination (βarr2-CLB,ΔCT) abolished the endocytic activity of β-arrestin-2 for all the GPCRs (*Figure 4—figure supplement 1a–f*). This indicates that the CT and CLB are sufficient to fully account for the GPCR-triggered endocytic activity of β-arrestin-2. Accordingly, we used the effect of mutating

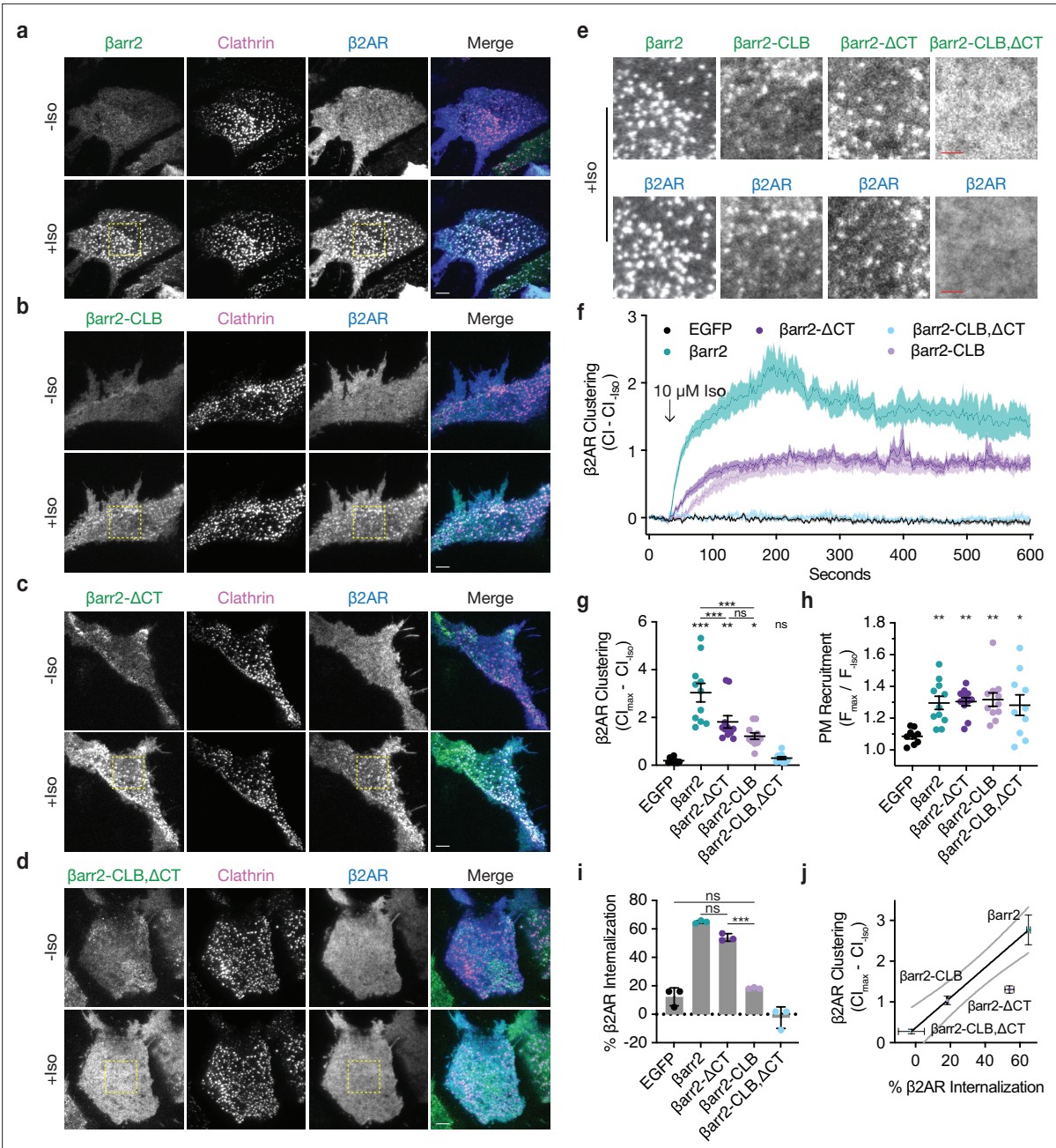

**Figure 3.** βarr2 C-terminus (CT) is not sufficient for β2-adrenergic receptor (β2AR) internalization. Representative live-cell total internal reflection fluorescence (TIRF) microscopy images of βarr1/2 double knockout HEK293s coexpressing clathrin-light-chain-DsRed (magenta) and FLAG-tagged β2AR (blue) with either EGFP-tagged βarr2 (**a**), β arr2-CLB (**b**), βarr2-ΔCT (**c**), or βarr2-CLB,ΔCT (**d**) (all in green) pre- and post-stimulation with 10 μM isoproterenol (Iso). Scale bars represent 5 μm. EGFP condition not shown (see *Figure 1a* for example). (**e**) Zoomed images corresponding to dashed boxes in panels a–d for βarr2 and β2AR images. Scale bars (red) represent 2.5 μm. (**f**) β2AR clustering index (CI, see Methods) pre- and post-stimulation with 10 μM Iso over 10 min. (**g**) Max plasma membrane recruitment of the indicated EGFP-tagged proteins in response to treatment with 10 μM Iso. (**h**) Max clustering index (CI) of β2AR calculated from within the first 300 s of (**f**) and normalized to clustering index prior to Iso treatment. For (**f–h**), data shown as mean ± standard error of the mean (SEM) ($n \geq 9$ cells, represented as dots in g and h). (**i**) Internalization of β2AR when coexpressed with the indicated EGFP-tagged proteins ($n = 3$, each dot is an average of three technical replicates) in βarr1/2 DKO HEK293 cells. (**j**) Correlation between β2AR clustering and internalization. Solid line is a simple linear regression fit to βarr2 and βarr2-CLB,ΔCT ($R^2 = 0.69$, dashed lines = 95% CI, vertical error = SEM, and horizontal error = std. dev.). For (**g–i**), significance was determined by ordinary one-way analysis of variance (ANOVA) (df = 4 for all, $F$ = 21.32, 4.828, and 117.6, respectively) with Tukey's test for multiple comparisons (ns $p \geq 0.05$, *$p < 0.05$, **$p < 0.01$, ***$p < 0.001$). All data shown are from at least three independent experiments.

The online version of this article includes the following source data and figure supplement(s) for figure 3:

*Figure 3 continued on next page*

*Figure 3 continued*

**Source data 1.** Source data for the figures displayed in *Figure 3*.

**Figure supplement 1.** βarr2 and mutants bind scFv30, βarr2-CLB,ΔCT acts as a dominant negative, CLB is required for β2-adrenergic receptor (β2AR) internalization in βarr1, and mutating a conserved residue (N218) in βarr2 abolished β2AR internalization.

**Figure supplement 1—source data 1.** Source data for results graphed in *Figure 3—figure supplement 1*.

each determinant individually to estimate the relative contribution of the other (non-mutated) determinant (*Figure 4b*).

A roughly linear relationship between the relative contribution of each determinant was observed for all the naturally occurring GPCRs included in our panel, with the contribution of both determinants summing to approximately 100% (*Figure 4c*). When compared in this way, receptors clustered according to differences in overall stability, as defined by the 'class A/B' classification scheme. The class A GPCRs (β2AR, κOR, μOR), defined by binding β-arrestins relatively weakly or transiently, were found to primarily utilize the CLB with little utilization of the CT. The class B GPCRs (V2R, M2R, NTSR1), defined by binding β-arrestins more strongly or stably, utilized both determinants in an additive manner.

The β2V2R chimera is considered a class B GPCR due to the V2R-derived cytoplasmic tail conferring enhanced binding to β-arrestin. This chimeric receptor also utilized both the CLB and CT for endocytosis, providing further support for the concept that the overall strength of GPCR/β-arrestin-binding impacts the utilization of endocytic determinants. Unlike the other receptors, however, the β2V2R departed from the linear relationship because the estimated contribution of each determinant summed to more than 100%. This suggests that this engineered GPCR can strongly utilize either the CT or CLB for endocytosis in a semi-redundant manner.

We further noted that, when compared to the β2AR, the β2V2R departed from the unity line primarily due to a stronger ability to use the β-arrestin CT for endocytosis (*Figure 4c*). This is interesting because the β2V2R differs from the wild-type β2AR mainly in its tail interaction with β-arrestin mediated by V2R-derived tail sequence. Relative to the V2R, the β2V2R departed from the linear

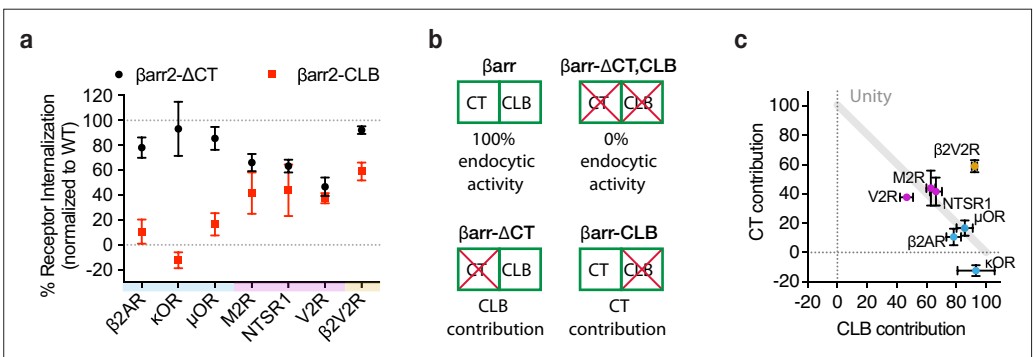

**Figure 4.** G-protein-coupled receptors (GPCRs) selectivity utilizes the βarr2 CLB and C-terminus (CT) for endocytosis. (**a**) Internalization of the CT (black) and CLB (red) mutants normalized to wild-type βarr2 for each receptor after 30 min of agonist (see *Figure 4—figure supplement 1*). Each dot is the mean of three independent experiments ± standard deviation. Shading indicates whether receptors are naturally occurring 'class a' (blue), 'class b' (magenta), or 'engineered class b' (gold). (**b**) Schematic summarizing the conceptual basis for estimating contributions of the CT and CLB. Contribution of each determinant within βarr2 is defined by subtracting internalization measured in the negative control (EGFP) from βarr2, βarr2-ΔCT, βarr2-CLB, and dividing the resulting values by control (EGFP) subtracted wild-type (βarr2) value. (**c**) Contribution to total endocytic activity of each determinant plotted as *x* and *y* coordinates for each receptor from panel (**a**). Unity is defined as 100% endocytic activity when individual activities are summed. Dot color corresponds to the typology described for panel (**a**). All data shown are from three independent experiments that were performed in βarr1/2 DKO HEK293 cells.

The online version of this article includes the following source data and figure supplement(s) for figure 4:

**Source data 1.** Source data for results graphed in *Figure 4*.

**Figure supplement 1.** Internalization of G-protein-coupled receptors (GPCRs) coexpressed with βarr2 wild type or mutants.

relationship primarily due to stronger utilization of the β-arrestin CLB. Moreover, the β2V2R is thought to differ from the wild-type V2R primarily in its core interaction with β-arrestin. Accordingly, we speculate that GPCRs differ in their ability to utilize the CT or CLB for endocytosis according to differences in the interactions that they make with β-arrestin through the receptor tail and core, respectively. This is likely an oversimplification considering the diversity of GPCRs β-arrestins engage. For example, the M2R was similar to other class B GPCRs in the present analysis, but its critical phosphorylation sites reside in the third intracellular loop (*Nakata et al., 1994*).

## Each endocytic determinant is oppositely coupled to the net signaling output of endogenous β2ARs

Having observed that differences in the GPCR/β-arrestin complex affect utilization of the CT relative to CLB, we next asked if each discrete endocytic determinant conversely influences the GPCR/β-arrestin interaction. To test this, we returned to the β2AR as a prototype and investigated how mutating the CT or CLB affects β2AR/βarr2 complex formation in living cells.

We first assessed complex formation using a NanoBiT real-time protein complementation assay. We observed no Iso-dependent response when interaction was measured between β2AR and Nb33, a biosensor that we consider a negative control because it recognizes activated μOR but not the β2AR (*Stoeber et al., 2018*). Iso produced a concentration-dependent increase in the interaction between the β2AR and wild-type β-arrestin-2. Similar results were obtained after mutating the CT and CLB individually or in combination, verifying that neither determinant is required for receptor/β-arrestin complex formation. However, we noted a trend suggesting additional specificity: Mutating the β-arrestin-2 CLB (βarr2-CLB) appeared to decrease the agonist potency (reported as $logEC_{50} \pm 95\%$ CI) for recruitment ($-6.8 \pm 0.1$), whereas mutating the CT (βarr2-ΔCT) increased it when compared to wild type ($-7.3 \pm 0.2$ and $-7.1 \pm 0.1$, respectively), and mutating both determinants in combination (βarr2-CLB,ΔCT) trended toward an intermediate effect ($-7.1 \pm 0.3$) (*Figure 5a*). We also noted that single mutation of the CLB slowed β2AR/βarr2 association relative to that observed for wild-type protein ($Tau_{95\% CI}$ = 81–108 s and 23–56, respectively), whereas the double (CLB and CT) mutation produced kinetics indistinguishable from wild type ($Tau_{95\% CI}$ = 31–37 s, *Figure 5b*). These results indicate that neither the CT nor CLB is required for β-arrestin to associate with receptors but suggest that each determinant influences the interaction in different ways.

More pronounced differences were observed when we assessed β2AR/β-arrestin complex formation functionally by assessing desensitization of the endogenous β2AR-elicited cAMP response. We verified a lack of β-arrestin-mediated desensitization in βarr1/2 DKO cells, as indicated by persistently elevated cAMP in the continued presence of agonist, and rescue of desensitization by recombinant wild-type βarr2 (*Figure 5c, d*). We also observed rescue of desensitization by the βarr2-CLB,ΔCT double mutant construct, despite this construct having no detectable endocytic activity, as well as rescue after single mutation of the β-arrestin CT. Remarkably, individually mutating the CLB severely impaired the ability of βarr2 to rescue the desensitization response (*Figure 5c, d*). Moreover, this signaling effect was phenocopied by making the same single-residue exchange (N218T) in the otherwise wild-type CLB (*Figure 5—figure supplement 1a, b*). Together, these results suggest that the discrete endocytic determinants in the β-arrestin CLB and CT, while not known to directly contact the receptor and not essential for GPCR/β-arrestin complex formation, are each coupled to the receptor/β-arrestin interface and produce opposing effects on the net signaling output of receptors.

## Discussion

Prior to this work, the ability of GPCRs to trigger the endocytic adaptor activity of β-arrestin was assumed to rely entirely on release of the β-arrestin CT. Revising this view, we show that the β-arrestin CT, while clearly capable of promoting GPCR endocytosis, is not necessary for this process. The CT-independent endocytic activity of β-arrestin is clearly evident in a β-arrestin knockout background. However, we note that a previous study using COS-1 cells, which naturally express β-arrestin at a low level, also reported CT-independent endocytosis (*Orsini and Benovic, 1998*). We define the CLB as a discrete determinant in β-arrestin that is critical for the endocytic activity of β-arrestin, and which can operate in the absence of the CT. We further show that the relative contribution of the β-arrestin CT and CLB in mediating the endocytic activity of β-arrestin varies depending on the GPCR bound,

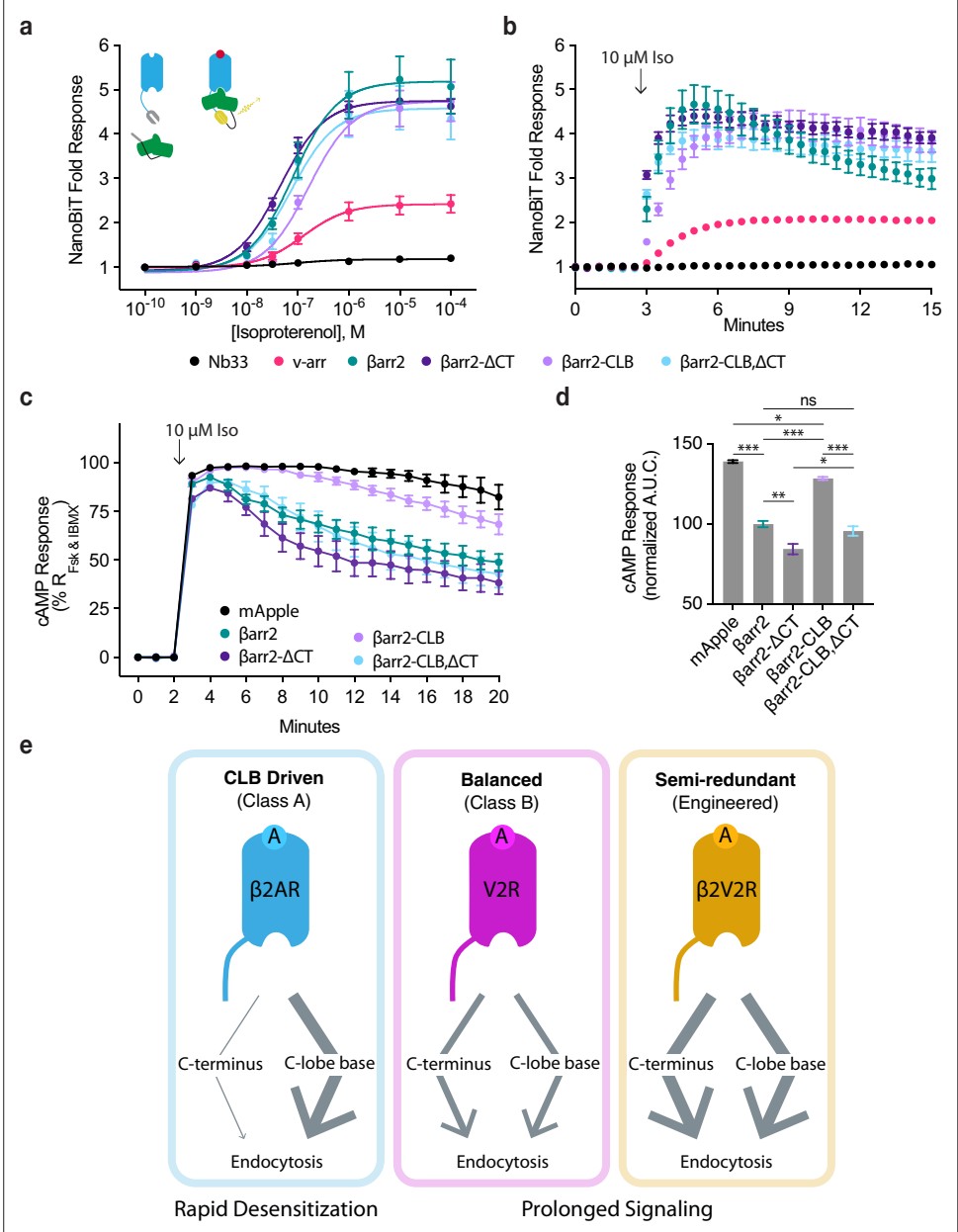

**Figure 5.** CLB and C-terminus (CT) determinants reveal two allosteric paths from G-protein-coupled receptors (GPCRs) to the endocytic network. Direct NanoBiT luciferase complementation of β2-adrenergic receptor (β2AR)-LgBiT and SmBiT-tagged: Nb33 (a μOR receptor-specific nanobody, black), visual arrestin (pink), wild-type β-arrestin-2 (green), CT mutant (dark purple), CLB mutant (light purple), or double mutant (cyan) measured as an end point across a range of isoproterenol (Iso) concentrations (**a**) and kinetically (**b**) pre- and post-stimulation with 10 μM Iso. Dose–response curves were generated with three-parameter nonlinear fit ($R^2$ = 0.94–0.99). (**c**) Endogenous β2AR cAMP response after stimulation with 10 μM Iso measured by a genetically encoded fluorescent cAMP biosensor, cADDis, and normalized to the response elicited by simultaneous treatment with 10 μM forskolin (Fsk) and 300 μM 3-isobutyl-1-methylxanthine (IBMX). (**d**) Area under the curve calculated from panel (**c**). All data are shown as mean ± standard error of the mean (SEM). from three independent experiments performed in βarr1/2 DKO HEK293 cells. Significance was determined by an ordinary one-way analysis of variance (ANOVA) (df = 4, $F$ = 112.1) with Tukey's multiple comparisons test. ns $p \geq 0.05$, *$p < 0.05$, ** $p < 0.01$, ***$p < 0.001$. (**e**) Diagram of proposed model involving two differentially utilized allosteric paths from GPCRs through β-arrestins to promote endocytosis. Class A GPCRs (blue), exemplified by the β2AR, primarily utilize the CLB to drive endocytosis while Class B GPCRs, exemplified by V2R (magenta) and β2V2R (gold) utilize both determinants. Arrows represent the proposed allosteric paths linking the GPCR/β-arrestin interface to the β-arrestin/clathrin-coated pit (CCP) interface,

*Figure 5 continued on next page*

*Figure 5 continued*

explaining how the CLB-dependent endocytic mode is coupled to rapid desensitization of receptor signaling while the CT-dependent mode enables prolonged signaling.

The online version of this article includes the following source data and figure supplement(s) for figure 5:

**Source data 1.** Source data for results graphed in *Figure 5*.

**Figure supplement 1.** N218 in βarr2 is required for endogenous β2-adrenergic receptor (β2AR) desensitization.

**Figure supplement 1—source data 1.** Source data for results graphed in *Figure 5—figure supplement 1*.

with class A receptors primarily dependent on the CLB and class B GPCRs dependent on the CT and CLB to a similar degree. By focusing on the β2AR as a prototypic class A GPCR, we additionally show that the β-arrestin CT can promote receptor clustering into CCPs despite having little ability to drive the subsequent endocytosis of receptors. Moreover, we show that mutations which selectively disrupt each of the endocytic determinants produce opposing effects on the net signaling output of receptors. In sum, these results support a model in which the endocytic activity of β-arrestins is triggered flexibly by GPCRs through distinct (CT and CLB-dependent) biochemical modes that differentially attenuate or prolong receptor signaling (*Figure 5e*).

Only two β-arrestins regulate hundreds of GPCRs and it is increasingly clear that β-arrestins function as highly flexible regulators of GPCR signaling, with the ability to produce a variety of downstream functional effects depending on receptor-specific differences communicated across the GPCR/β-arrestin interface. The present results extend this concept to flexibility and diversity at the β-arrestin/CCP interface. Our finding that mutations which disrupt either the CT or CLB produce different effects on endogenous β2AR signaling suggest that each endocytic mode, in addition to being triggered in a GPCR-specific manner, is coupled to opposing effects on the net signaling output of receptors.

Beyond providing new insight into the cell biology of GPCRs and β-arrestins, we believe that our results have implications for understanding endocytic adaptor functions more generally. The best studied among these is AP2, a protein complex that is essential for CCP nucleation and co-assembles with clathrin. AP2 switches from a 'closed' to 'open' state triggered by the binding of PIP2 and endocytic cargo (*Kovtun et al., 2020*). β-Arrestins are structurally distinct from AP2 and also differ functionally, as they are not required for CCP formation and associate with CCPs primarily after assembly (*Puthenveedu and von Zastrow, 2006*; *Santini et al., 2002*). Nevertheless, β-arrestins also switch from 'closed' to 'open' states triggered by binding GPCR cargo and aided, in many cases, by the binding of PIP2 (*Gaidarov et al., 1999*; *Huang et al., 2020*; *Kang et al., 2014*; *Shukla et al., 2013*). The present results provide evidence that β-arrestin can switch into more than one active or 'open' state, and do so selectively depending on the GPCR bound. To our knowledge, these findings provide the first example of an endocytic adaptor protein capable of undergoing cargo-specific mode switching. The existence of such a flexible relationship between cargo binding and functional activation of the adaptor protein suggests that adaptor proteins have the capacity to communicate more nuanced information to the CCP than the simple presence or absence of a cognate cargo. In addition, these findings raise multiple new questions about the biochemical and structural basis for the discrete active or 'open' states of β-arrestin that we resolve here according to functional differences.

## Ideas and speculation

While the present results provide strong evidence for the existence of discrete endocytic modes differing in dependence on the β-arrestin CT and CLB, they leave unresolved the biochemical details of the latter mode. In contrast to the β-arrestin CT, for which relevant CCP-associated interacting partners are well known, how the β-arrestin CLB engages CCPs remains to be determined. We show that the discrete function of the CLB depends on engagement with partners other than the clathrin terminal domain and the β-appendage of AP2, which are key domains interacting with the β-arrestin CT. We note that many proteins are presently known to bind AP2 or the clathrin terminal domain (*Miele et al., 2004*; *Muenzner et al., 2017*; *Willox and Royle, 2012*). We also note that the relatively understudied C-lobe engages multiple additional proteins and that two of these, PIP5K1A (*Jung et al., 2021*; *Nelson et al., 2008*) and PDE4D5 (*Baillie et al., 2007*; *Perry et al., 2002*), have binding sites which overlap the CLB and are already known to affect endocytosis and GPCR desensitization. However, neither protein is currently known to be enriched in CCPs, which is presumably required

for the CLB to drive GPCR-triggered accumulation in CCPs without the CT as we show. Moreover, the CCP lattice associates with more than 50 additional proteins, some only transiently during CCP maturation, and a number with known regulatory effects (*Kirchhausen et al., 2014*; *Merrifield and Kaksonen, 2014*; *Mettlen et al., 2018*; *Traub, 2011*). Thus, further study will be required to determine whether and how these proteins, and/or other candidates yet to be identified, mediate the signal modulating and trafficking functions of the CLB delineated here.

## Materials and methods

### Cell culture, expression constructs, and transfections

Parental and βarr1/2 double knockout HEK 293A cell lines, generously provided by Asuka Inouye and Silvio Gutkind (*O'Hayre et al., 2017*), were cultured in complete growth Dulbecco's modified Eagle's medium (DMEM, Life Technologies, 11965118) supplemented with 10% fetal bovine serum (UCSF Cell Culture Facility). Cell cultures were verified to be free of mycoplasma contamination by enzymatic assay (MycoAlert, Lonza, LT07-318). DNA transfections were carried out using Lipofectamine 2000 (Thermo, 11668019) according to the manufacturer's protocol. Cells were transfected 24–48 hr before experiments.

All receptor constructs were N-terminally FLAG tagged. The human β2AR, V2R, μOR, and κOR were previously described (*Cao et al., 1999*; *Chu et al., 1997*; *Tanowitz and von Zastrow, 2003*). NTSR1 was a generous gift from Brian Kobilka (*Huang et al., 2020*). The β2AR–V2R chimera was a generous gift from Marc Caron (*Oakley et al., 1999*). M2R was previously generated in the lab by James Hislop. The β2AR-3S was previously described (*Hausdorff et al., 1991*). β2AR-LgBiT was generated by PCR amplification from the LgBiT Expression Vector (Promega) and inserted into PCR linearized FLAG-tagged β2AR using InFusion HD.

β-Arrestin-2–GFP and β-arrestin-2–mApple were previously described (*Barak et al., 1997*; *Eichel et al., 2016*). β-Arrestin-2-CCS–EGFP was previously described (*Eichel et al., 2018*). All EGFP- and mApple-tagged βarr2 CLB and ΔCT mutants were generated by PCR amplifying from βarr2-EGFP and subcloning into EGFP-N1 (ClonTech) or mApple-N1 (*Steinbach et al., 2008*) using InFusion HD (Takara Bio). Point mutants were generated by site directed mutagenesis PCR to generate linear DNA that was then closed using InFusion HD. β-Arrestin-2-SmBiT constructs were generated by removing EGFP from βarr2, βarr2-ΔCT, βarr2-CLB, and βarr2-ΔCT,CLB constructs by digestion with ApaI and XbaI followed by insertion of SmBiT by two rounds of PCR followed by InFusion HD.

β-Arrestin-1–EGFP was generated by PCR amplifying from β-arrestin-1–mVenus, which was a gift from R. Sunahara (University of California, San Diego), and subcloned into EGFP-N1. All βarr1 mutants were generated by PCR amplifying from βarr1-EGFP and subcloning into EGFP-N1 using InFusion HD.

EGFP-tagged visual arrestin (v-arr) was generated by synthesis of bovine visual arrestin (Twist Biosciences) and subcloned into EGFP-N1 using InFusion.

Visual arrestin and β-arrestin-2 EGFP-tagged chimeras were generated by synthesis of the template chimera, ChiA (Twist Bioscience), and subcloned into EGFP-N1. Subsequent chimeras, ChiA.1–14, were generated by PCR and InFusion HD (Takara Bio).

scFv30-LgBiT was generated by PCR amplifying scFv30 from scFv30-YFP, a generous gift Arun Shukla, and LgBiT from LgBiT Expression Vector (Promega) and inserted sequentially into pCDNA3.1 by InFusion HD.

Clathrin–dsRed was previously described (*Merrifield et al., 2002*).

Nb33-SmBiT was previously generated in the von Zastrow lab by Joy Li.

GST-tagged clathrin terminal domain (1–363) and the β-appendage of AP2 (701–937) were generous gifts from Jeffrey Benovic and Harvey McMahon (*Kang et al., 2009*).

See *Supplementary file 1* for all primers used in this study.

### Live-cell TIRF microscopy imaging

TIRF microscopy was performed at 37°C using a Nikon Ti-E inverted microscope equipped for through-the-objective TIRF microscopy and outfitted with a temperature-, humidity-, and $CO_2$-controlled chamber (Okolab). Images were obtained with an Apo TIRF ×100, 1.49 numerical aperture objective (Nikon) with solid-state 405, 488, 561, and 647 nm lasers (Keysight Technologies). An Andor iXon DU897 EMCCD camera controlled by NIS-Elements 4.1 software was used to acquire

image sequences every 2 s for 10 min. βarr1/2 double knockout HEK293s were transfected as indicated according to the manufacturer's protocol 48 hr before imaging and then plated on poly-L-lysine (0.0001%, Sigma) coated 35-mm glass-bottom culture dishes (MatTek Corporation) 24 hr before imaging. Cells were labeled with monoclonal FLAG antibody (M1) (1:1000, Sigma F-3040) conjugated to Alexa Fluor 647 dye (Life Technologies) for 10 min at 37 °C before imaging, washed, and imaged live in DMEM without phenol red (UCSF Cell Culture Facility) supplemented with 30 mM HEPES [N-(2-hydroxyethyl)piperazine-N'-(2-ethanesulfonic acid)], pH 7.4 (UCSF Cell Culture Facility). Cells were treated by bath application of isoproterenol at the indicated time. At least three independent experiments were performed for all live-cell microscopy.

## TIRF microscopy image analysis

Quantitative image analysis was performed on unprocessed images using ImageJ and Fiji software (*Schindelin et al., 2012*; *Schneider et al., 2012*). To quantify change in β-arrestin fluorescence over time in TIRF microscopy images, which was reported as plasma membrane recruitment, fluorescence values were measured over the entire time series in a region of interest (ROI) corresponding to the cell. Fluorescence values of the ROI were normalized to fluorescence values before agonist addition. Minimal bleed-through and photobleaching was verified using single-labeled and untreated samples, respectively. Line scan analysis of receptor, β-arrestin, or clathrin fluorescence was carried out using the Fiji plot profile function to measure pixel values from the five pixel wide lines shown. Clustering index was determined using the skew statistical measurement applied to fluorescence intensity values of M1-Alexa647 labeled receptor pixels in a ROI corresponding to the cell as has been previously described (*Eichel et al., 2018*).

## Flow cytometry internalization assays

All internalization assays were performed with βarr1/2 DKO HEK293 cells that were transfected in 6 cm dishes according to the manufacturer's protocol 24 hr before beginning the assay with 1000 ng of GPCR DNA paired with 400–800 ng of βarr (or one of its mutants) DNA. At least 3 hr before drug treatment, cells were lifted using TrypLE Express (Thermo Scientific), a dissociation reagent that leaves extracellular epitopes intact, resuspended in complete media, transferred to 12-well plates in triplicate, and incubated under standard culture conditions. Cells were then treated with the corresponding agonist (see figure legends) for the indicated period and placed on ice to stop trafficking. Cells were washed once with ice-cold phosphate-buffered saline (PBS) supplemented with 2 mM $CaCl_2$ followed by labeling of FLAG-tagged surface receptors with M1 antibody (Sigma Aldrich, F3040) conjugated to Alexa Fluor 647 (Thermo Scientific) for 30 min at 4 °C while gently shaking. Surface staining of receptors was measured using a CytoFlex (Beckman Coulter) with gates set for single cells expressing EGFP (see *Figure 1—source data 1*). Examination of unstimulated cells verified that fluorescence from either EGFP or surface receptors was similar across conditions within each GPCR tested. Percent internalization was calculated by taking the mean M1-647 fluorescence for the agonist treated cells and dividing it by the same measure for the corresponding unstimulated cells, subtracted from one, and multiplied by 100. At least three independent experiments were performed for all internalization assays.

## siRNA transfection

Cells were seeded at 30% confluency in a 6 cm dish and transfected overnight with 3 µl of 20 µM stocks of either AllStars Negative Control (Qiagen, Germantown, MD; 1027281) or CHC17 (5'- AAGC AATGAGCTGTTTGAAGA-3') siRNA with RNAiMax Lipofectamine (Invitrogen) according to the manufacturer's protocol. Media was changed the next day and cells recovered for another 48 hr before experimentation. If DNA transfection was also required, this was done using Lipofectamine 2000 24–48 hr before experimentation.

## V2R phosphopeptide

The V2Rpp peptide (ARGRpTPPpSLGPQDEpSCpTpTApSpSpSLAKDTSS) was obtained by custom peptide synthesis (Tufts University Core Facility).

## Purification of GST-tagged proteins

Production of GST-clathrin terminal domain and GST-AP2 β-appendage proteins was initiated by transformation of *E. coli* BL21 DE3 cells, transformed with either GST fusion protein construct. Starter

cultures were grown at 37°C overnight in Luria Broth and used to inoculate Terrific Broth. Cultures were grown at 37°C until OD600 reached 0.6 and then induced with 300 µM isopropyl-1-thio-β-D-galactopyranoside for 20 hr at 20°C. Cultures were centrifuged for 30 min at 4000 × *g* and frozen at −80°C. Pellets were resuspended in lysis buffer (50 mM HEPES pH 7.4, 500 mM NaCl, 10 µM leupeptin, benzonase, and lysozyme) and lysed by sonication. Lysate was clarified by centrifugation at 37,800 × *g* for 30 min at 4°C and then applied to glutathione agarose (Thermo Fisher, PI16100). Agarose was washed with 20 column volumes of lysis buffer followed by 20 column volumes of wash buffer (20 mM HEPES pH 7.4, 150 mM NaCl, 2 mM dithiothreitol [DTT]) and then protein was eluted by supplementing the wash buffer with 10 mM reduced glutathione. Buffer was exchanged to freeze buffer (20 mM HEPES 7.4, 150 mM NaCl, 2 mM DTT, and 20% glycerol), protein was flash frozen in LN2, and stored at −80°C until use.

## GST pull-downs

Purified GST-tagged protein (1 mg) was diluted to 500 µL in protein buffer (20 mM HEPES 7.4, 150 mM NaCl, 2 mM DTT), applied to 50 µl of settled glutathione agarose, batch bound while rotating for 1.5 hr at room temperature, and then washed 3× with protein buffer. β-Arrestin-1/2 double knockout cells transiently expressing either βarr2-EGFP or βarr2-ΔCT-EGFP were harvested from 10 cm dishes, lysed by resuspension in 1 ml of cold lysis buffer (20 mM HEPES [N-(2-hydroxyethyl)piperazine-N′-(2-ethanesulfonic acid)] pH 7.4, 150 mM NaCl, 2 mM DTT [dithiothreitol], 5 mM EDTA [ethylenedi-aminetetraacetic acid], 1% NP-40) containing Complete Mini, EDTA-free (Roche, Mannheim, Germany; 11836170001) and PhosSTOP (Roche, 04906837001), clarified by centrifugation for 30 min at 17,000 × *g*, and applied to glutathione agarose that was prebound to GST-tagged proteins. All samples were supplemented with 10 µM V2Rpp and incubated on a rotator for 3 hr at 4°C. Glutathione agarose was then washed 4× with lysis buffer and protein was eluted by application of lysis buffer supplemented with 10 mM reduced glutathione.

Protein samples were prepared for analysis by adding NuPage LDS Sample Buffer (Invitrogen, NP0007) containing 15 mM DTT (Sigma, D0632) and heating at 85°C for 5 min. Samples were loaded into a NuPAGE 4–12% Bis-Tris Protein gels (Invitrogen, NP0335BOX) and either stained with Instant-Blue Coomassie (Abcam, ab119211) or transferred to nitrocellulose (Bio-Rad, 162-0112).

After transfer onto nitrocellulose, membranes were stained with Ponceau S solution (Sigma, P7170) to check transfer quality. Membranes were washed with TBS to remove the Ponceau S solution before blocking with Odyssey TBS blocking buffer (LI-COR Biosciences, Lincoln, NE; 927-50000). Primary antibodies were used to probe for proteins of interest with the corresponding secondary antibodies. The blots were imaged on an Odyssey Infrared Imaging System (LI-COR Biosciences). Scans of uncropped and unprocessed blots are provided in *Figure 1—figure supplement 2*.

## NanoBiT complementation

β-Arrestin-1/2 double knockout HEK293s were plated in 6-well dishes, transfected with β2AR-LgBiT (200 ng per well of a 6-well plate) paired with one of the following: Nb33-EGFP-SmBiT, arr1-SmBiT, βarr2-SmBiT, βarr2-ΔCT-SmBiT, βarr2-CLB-SmBiT, or βarr2-CLB,ΔCT-SmBiT (50 ng per well of a 90% confluent 6-well plate). Twenty-four hours later, cells were lifted with TrpLE Express, resuspended in 37°C assay buffer (135 mM NaCl, 5 mM KCl, 0.4 mM MgCl$_2$, 1.8 mM CaCl$_2$, 20 mM HEPES, and 5 mM D-glucose, adjusted to pH 7.4), and transferred to a white flat bottom 96-well plate in triplicate with 20,000 cells per well. Coelenterazine-H (Thermo Scientific) in assay buffer prewarmed to 37°C was added to a final concentration of 5 µM and incubated for at least 5 min before data collection. For kinetic experiments, three time points were collected to establish baseline before vehicle or 10 µM isoproterenol addition (both of which contained 5 µM coelenterazine-H). Fold response was calculated by averaging the values across each triplicate and then dividing the isoproterenol treated samples by the corresponding buffer treated samples. For dose–response experiments, isoproterenol was added to the indicated final concentrations in a 37°C assay buffer and data were collected for 20 min.

For complementation with scFv30-LgBiT, β-arrestin-1/2 double knockout HEK293s were plated in 10 cm dishes and transfected the next day with scFv30-LgBiT (3000 ng) paired with one of the following βarr2-SmBiT, βarr2-ΔCT-SmBiT, βarr2-CLB-SmBiT, or βarr2-CLB,ΔCT-SmBiT (1000 ng). After 24 hr, cells were lifted with PBS EDTA, spun down, lysed in 1 ml of buffer (50 mM HEPES 7.4, 150 mM NaCl, 1% NP-40, 5 mM EDTA) containing Complete Mini, EDTA-free (Roche, Mannheim, Germany;

11836170001) and PhosSTOP (Roche, 04906837001), and clarified by centrifugation for 30 min at 17,000 × $g$. Samples were transferred to a 96-well plate in duplicate, followed by addition of the indicated concentration of V2Rpp, and incubated for 15–20 min at room temperature. Coelenterazine-H was added to 5 µM and incubated for 5 min before data were collected on a plate reader (CLARIOStar Plus) for 30 min.

Fold response was calculated by averaging the values across each triplicate, or duplicate for scFv30-LgBiT, and then dividing the maximum by the minimum responses within each condition or dose range. Dose–response curves were generated by a three-parameter nonlinear fit and tau values for time series data were determined using a one-phase association, both were calculated in Prism 9. At least three independent experiments were performed for all NanoBiT assays.

## Live-cell cAMP assay

β-Arrestin-1/2 double knockout HEK293 cells were plated in 6-well plates, transfected with either mApple, βarr2-mApple, βarr2-ΔCT-mApple, βarr2-CLB-mApple, or βarr2-CLB,ΔCT-mApple. The following day, cells were lifted with TrypLE, transduced with CMV cADDis Green Upward cAMP sensor (Montana Molecular) according to the manufacturer's instructions, and transferred in triplicate at 50,000 cells per well in a black clear bottom 96-well plate (Corning). On the day of the experiment, the media was removed and replaced with 37°C assay buffer (135 mM NaCl, 5 mM KCl, 0.4 mM $MgCl_2$, 1.8 mM $CaCl_2$, 20 mM HEPES, and 5 mM D-glucose, adjusted to pH 7.4) and incubated for 5 min in the prewarmed plate reader (H4 Synergy BioTek). Similar expression of mApple-tagged plasmids was verified by fluorescence with monochromators set to Ex: 568/9.0 and Em: 592/13.5. Next, cADDis fluorescence baseline, and similarity of its expression across conditions, was established by three time points a minute apart using monochromators set to Ex: 500/9.0 and Em: 530/20.0. Isoproterenol was then added to a final concentration of 10 µM and cADDis fluorescence was measured every minute for the indicated time. At the end of the kinetic run 10 µM forskolin (Fsk) and 300 µM 3-isobutyl-1-methylxanthine (IBMX) were added. Percent cAMP response was calculated by subtracting the baseline fluorescence from each time point and then dividing by the max response after Fsk/IBMX addition. No fluorescence bleed through nor significant photobleaching were observed in separate experiments where cells expressing only mApple or cADDis green were measured with the same optical configuration. At least three independent experiments were performed.

## Sequence logos

PSI-BLAST was performed against bovine β-arrestin-2 protein sequence and iterated until convergence. Protein sequences annotated as β-arrestin and longer than 300 amino acids were kept while all others were discarded. Aligned regions of interest corresponding to 9 amino acids windows around either the N218 in the C-lobe base, the clathrin-binding motif, or the AP2β-binding motifs were extracted and were input into a previously logo generator (*Crooks et al., 2004*).

## Statistical analysis

Quantitative data are expressed as the mean and error bars represent the standard error of the mean (SEM) or standard deviation (SD) unless otherwise indicated. Scatter plots are overlaid with mean and SEM. Determination of statistical significance is described in each figure legend and was calculated using Prism 9.0 (GraphPad Software). *$p < 0.05$; **$p < 0.01$; ***$p < 0.001$ when compared with control, no-treatment, or other conditions. All experiments showing representative data were repeated at least three independent times with similar results. Independent experiments represent biological replicates.

## Acknowledgements

We thank J Benovic, M Bouvier, A Inoue, and S Gutkind for sharing reagents and discussions; B Kobilka, J Janetzko, S Sivaramakrishnan, A Frost, R Edwards, R D Mullins, M Thompson, E Blythe, as well as other von Zastrow laboratory and Manglik laboratory members for discussions. All live-cell

imaging experiments were performed in the UCSF Center for Advanced Light Microscopy directed by D Larsen. These studies were supported by grants from the U.S. National Institutes of Health R01DA010711 and R01DA012864 (MvZ) and DP5OD023048 (AM). BBR is a recipient of an American Heart Association Predoctoral Fellowship (19PRE34380570).

## Additional information

### Funding

| Funder | Grant reference number | Author |
|---|---|---|
| NIH Office of the Director | DP5OD023048 | Aashish Manglik |
| National Institutes of Health | R01DA010711 | Mark von Zastrow |
| National Institutes of Health | R01DA012864 | Mark von Zastrow |
| American Heart Association | 19PRE34380570 | Benjamin Barsi-Rhyne |

The funders had no role in study design, data collection, and interpretation, or the decision to submit the work for publication.

### Author contributions
Benjamin Barsi-Rhyne, Conceptualization, Data curation, Formal analysis, Funding acquisition, Investigation, Visualization, Methodology, Writing - original draft, Writing - review and editing; Aashish Manglik, Conceptualization, Resources, Supervision, Funding acquisition, Project administration, Writing - review and editing; Mark von Zastrow, Conceptualization, Resources, Supervision, Funding acquisition, Writing - original draft, Project administration, Writing - review and editing

### Author ORCIDs
Benjamin Barsi-Rhyne http://orcid.org/0000-0002-6610-1766
Aashish Manglik http://orcid.org/0000-0002-7173-3741
Mark von Zastrow http://orcid.org/0000-0003-1375-6926

### Decision letter and Author response
Decision letter https://doi.org/10.7554/eLife.81563.sa1
Author response https://doi.org/10.7554/eLife.81563.sa2

## Additional files

### Supplementary files
- MDAR checklist
- Supplementary file 1. List of primers used.

### Data availability
All numerical data used to generate the figures have been included in the supporting data file. Source data for each figure panels are included as a separate worksheet in the combined excel document.

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

# Appendix 1

## Appendix 1—key resources table

| Reagent type (species) or resource | Designation | Source or reference | Identifiers | Additional information |
|---|---|---|---|---|
| Cell line (*Homo sapiens*) | Parental HEK293 | ***O'Hayre et al., 2017*** (PMID: 28634209) | | |
| Cell line (*Homo sapiens*) | HEK293 βarr1/2 double knockout, line 1 | ***O'Hayre et al., 2017*** (PMID: 28634209) | | |
| Cell line (*Homo sapiens*) | HEK293 βarr1/2 double knockout, line 2 | ***O'Hayre et al., 2017*** (PMID: 28634209) | | |
| Recombinant DNA reagent | FLAG-β2AR (plasmid) | ***Cao et al., 1999*** (PMID: 10499588) | | |
| Recombinant DNA reagent | FLAG-β2AR-3S (plasmid) | ***Hausdorff et al., 1991*** (PMID: 1849641) | | |
| Recombinant DNA reagent | FLAG-V2R (plasmid) | Lefkowitz Laboratory, Duke University | | |
| Recombinant DNA reagent | FLAG-µOR (plasmid) | ***Tanowitz and von Zastrow, 2003*** (PMID: 12939277) | | |
| Recombinant DNA reagent | FLAG-κOR (plasmid) | ***Chu et al., 1997*** (PMID: 9341153) | | |
| Recombinant DNA reagent | FLAG-NTSR1 (plasmid) | ***Huang et al., 2020*** (PMID: 31945771) | | |
| Recombinant DNA reagent | FLAG-β2V2R (plasmid) | ***Oakley et al., 1999*** (PMID: 10542263) | | |
| Recombinant DNA reagent | FLAG-M2R (plasmid) | This paper | | See Materials and methods |
| Recombinant DNA reagent | FLAG-β2AR-LgBiT (plasmid) | This paper | | See Materials and methods |
| Recombinant DNA reagent | βarr2-EGFP (plasmid) | ***Barak et al., 1997*** (PMID: 9346876) | | |
| Recombinant DNA reagent | βarr2-mApple (plasmid) | ***Eichel et al., 2016*** (PMID: 26829388) | | |
| Recombinant DNA reagent | Clathrin-light-chain-dsRed (plasmid) | ***Merrifield et al., 2002*** (PMID: 12198492) | | |
| Recombinant DNA reagent | βarr2-CCS-EGFP (plasmid) | ***Eichel et al., 2018*** (PMID: 29720660) | | |
| Recombinant DNA reagent | βarr2-ΔCT-EGFP (plasmid) | This paper | | See Materials and methods |
| Recombinant DNA reagent | v-arr-EGFP (plasmid) | This paper | | See Materials and methods |
| Recombinant DNA reagent | ChiA-EGFP (plasmid) | This paper | | See Materials and methods |
| Recombinant DNA reagent | ChiA.1-EGFP (plasmid) | This paper | | See Materials and methods |
| Recombinant DNA reagent | ChiA.2-EGFP (plasmid) | This paper | | See Materials and methods |
| Recombinant DNA reagent | ChiA.3-EGFP (plasmid) | This paper | | See Materials and methods |
| Recombinant DNA reagent | ChiA.4-EGFP (plasmid) | This paper | | See Materials and methods |
| Recombinant DNA reagent | ChiA.5-EGFP (plasmid) | This paper | | See Materials and methods |
| Recombinant DNA reagent | ChiA.6-EGFP (plasmid) | This paper | | See Materials and methods |
| Recombinant DNA reagent | ChiA.7-EGFP (plasmid) | This paper | | See Materials and methods |
| Recombinant DNA reagent | ChiA.8-EGFP (plasmid) | This paper | | See Materials and methods |

*Appendix 1 Continued on next page*

*Appendix 1 Continued*

| Reagent type (species) or resource | Designation | Source or reference | Identifiers | Additional information |
|---|---|---|---|---|
| Recombinant DNA reagent | ChiA.9-EGFP (plasmid) | This paper | | See Materials and methods |
| Recombinant DNA reagent | ChiA.10-EGFP (plasmid) | This paper | | See Materials and methods |
| Recombinant DNA reagent | ChiA.11-EGFP (plasmid) | This paper | | See Materials and methods |
| Recombinant DNA reagent | ChiA.12-EGFP (plasmid) | This paper | | See Materials and methods |
| Recombinant DNA reagent | ChiA.13-EGFP (plasmid) | This paper | | See Materials and methods |
| Recombinant DNA reagent | ChiA.14-EGFP (plasmid) | This paper | | See Materials and methods |
| Recombinant DNA reagent | βarr2-CLB-EGFP (plasmid) | This paper | | See Materials and methods |
| Recombinant DNA reagent | βarr2-CLB,ΔCT-EGFP (plasmid) | This paper | | See Materials and methods |
| Recombinant DNA reagent | EGFP-Nb33-SmBiT (plasmid) | This paper | | See Materials and methods |
| Recombinant DNA reagent | βarr2-SmBiT (plasmid) | This paper | | See Materials and methods |
| Recombinant DNA reagent | βarr2-ΔCT-SmBiT (plasmid) | This paper | | See Materials and methods |
| Recombinant DNA reagent | βarr2-CLB-SmBiT (plasmid) | This paper | | See Materials and methods |
| Recombinant DNA reagent | βarr2-CLB,ΔCT-SmBiT (plasmid) | This paper | | See Materials and methods |
| Recombinant DNA reagent | mApple (plasmid) | *Steinbach et al., 2008* (PMID: 18454154) | | |
| Recombinant DNA reagent | EGFP (plasmid) | Clontech | | Discontinued |
| Recombinant DNA reagent | βarr2-ΔCT-mApple (plasmid) | This paper | | See Materials and methods |
| Recombinant DNA reagent | βarr2-CLB-mApple (plasmid) | This paper | | See Materials and methods |
| Recombinant DNA reagent | βarr2-CLB,ΔCT-mApple (plasmid) | This paper | | See Materials and methods |
| Commercial assay or kit | In-Fusion HD Cloning | Takara | 638920 | |
| Recombinant DNA reagent | βarr1-ΔCT-EGFP (plasmid) | This paper | | See Materials and methods |
| Recombinant DNA reagent | GST-AP2β (plasmid) | Jeff Benovic and Harvey McMahon | | |
| Recombinant DNA reagent | GST-CTD (plasmid) | *Kang et al., 2009* (PMID: 19710023) | | |
| Strain, strain background (*Escherichia coli*) | BL21 DE3 | QB3 MacroLab UC Berkeley | | |
| Recombinant DNA reagent | scFv30-LgBiT (plasmid) | This paper | | See Materials and methods |
| Commercial assay, kit | Lipofectamine 2000 | Thermo Fisher Scientific | 11668019 | |
| Commercial assay, kit | Lipofectamine RNAi Max | Invitrogen | 13778075 | |
| Transfected construct (human) | siRNA, negative control | Qiagen | 1027281 | |
| Transfected construct (human) | siRNA, clathrin heavy chain | Qiagen | | 5'- AAGCAATGAGCTGTTTGAAGA-3' |

*Appendix 1 Continued on next page*

*Appendix 1 Continued*

| Reagent type (species) or resource | Designation | Source or reference | Identifiers | Additional information |
|---|---|---|---|---|
| Peptide, recombinant protein | V2Rpp | Tufts University Core Facility | | ARGRTPPSLGPQDESCTTASSSLAKDTSS (phosphorylated residues underlined) |
| Recombinant DNA reagent | βarr1-CLB-EGFP (plasmid) | This paper | | See Materials and methods |
| Recombinant DNA reagent | βarr1-ΔSL,ΔCT-EGFP (plasmid) | This paper | | See Materials and methods |
| Recombinant DNA reagent | βarr1-CLB,ΔCT-EGFP (plasmid) | This paper | | See Materials and methods |
| Recombinant DNA reagent | βarr1-CLB,ΔSL,ΔCT-EGFP (plasmid) | This paper | | See Materials and methods |
| Recombinant DNA reagent | βarr2-CLB,I208L,I215L-EGFP (plasmid) | This paper | | See Materials and methods |
| Recombinant DNA reagent | βarr2-D205S-EGFP (plasmid) | This paper | | See Materials and methods |
| Recombinant DNA reagent | βarr2-N216P-EGFP (plasmid) | This paper | | See Materials and methods |
| Recombinant DNA reagent | βarr2-N218T-EGFP (plasmid) | This paper | | See Materials and methods |
| Recombinant DNA reagent | βarr2-H220A-EGFP (plasmid) | This paper | | See Materials and methods |
| Recombinant DNA reagent | βarr2-L208I,L215I-EGFP (plasmid) | This paper | | See Materials and methods |
| Recombinant DNA reagent | βarr2-CLB,I208L,I215L-mApple (plasmid) | This paper | | See Materials and methods |
| Recombinant DNA reagent | βarr2-D205S-mApple (plasmid) | This paper | | See Materials and methods |
| Recombinant DNA reagent | βarr2-N216P-mApple (plasmid) | This paper | | See Materials and methods |
| Recombinant DNA reagent | βarr2-N218T-mApple | This paper | | See Materials and methods |
| Recombinant DNA reagent | βarr2-H220A-mApple (plasmid) | This paper | | See Materials and methods |
| Recombinant DNA reagent | βarr2-L208I,L215I-mApple (plasmid) | This paper | | See Materials and methods |
| Commercial assay, kit | cADDis Green Upward | Montana Molecular | #U0200G | |
| Chemical compound, drug | Forskolin (Fsk) | Sigma-Aldrich | F6886 | |
| Chemical compound, drug | 300 µM 3-isobutyl-1-methylxanthine (IBMX) | Sigma-Aldrich | F5879 | |
| Chemical compound, drug | (−)-Isoproterenol hydrochloride | Sigma-Aldrich | I6504 | |
| Chemical compound, drug | DADLE, [D-Ala2, N-Me-Phe4, Gly5-ol]-Enkephalin acetate salt | Sigma-Aldrich | E7131 | |
| Chemical compound, drug | DAMGO, [D-Ala2, N-Me-Phe4, Gly5-ol]-Enkephalin acetate salt | Sigma-Aldrich | E7384 | |
| Chemical compound, drug | Carbamoylcholine chloride, ≥98% (titration), crystalline | Sigma-Aldrich | C4382 | |
| Chemical compound, drug | Neurotensin, ≥90% (HPLC) | Sigma-Aldrich | N6383 | |
| Chemical compound, drug | AVP [Arg8]-Vasopressin acetate salt | Sigma-Aldrich | V9879 | |
| Chemical compound, drug | Coelenterazine-H | Thermo Fisher Scientific | 50-995-840 | |
| Antibody | αGFP (mouse diclonal) | Roche | 11814460001 | 1:1000 |

*Appendix 1 Continued on next page*

*Appendix 1 Continued*

| Reagent type (species) or resource | Designation | Source or reference | Identifiers | Additional information |
|---|---|---|---|---|
| Antibody | αGAPDH (rabbit monoclonal) | Cell Signalling Technologies | 5174S | 1:1000 |
| Antibody | M1 anti-flag (mouse monoclonal) | Sigma-Aldrich | F-3040 | 1:1000 |
| Antibody | Donkey Anti-Mouse IgG Antibody, IRDye 680RD Conjugated – 0.5 mg (donkey polyclonal) | Li-cor Biosciences | 926-68072 | 1:3000 |
| Antibody | IRDye 800CW Donkey anti-Rabbit IgG (H+L), 0.5 mg (donkey polyclonal) | Li-cor Biosciences | 926-32213 | 1:3000 |
| Commercial assay, kit | Alexa Fluor 647 Protein Labeling Kit | Thermo Fisher Scientific | A20173 | |
| Software, algorithm | Prism | GraphPad | | 9.0 |
| Software, algorithm | ImageJ | https://imagej.net/downloads | | 2.0.0-rc-54/1.51g |
| Software, algorithm | Excel | Microsoft | | 16.11.1 |
| Software, algorithm | ChimeraX | UCSF Resource for Biocomputing, Visualization, and Informatics | | 1.4 |
| Software, algorithm | Illustrator CC | Adobe | | 21.0.2 |
| Software, algorithm | Python | Python Software Foundation | | 3.7.4 |

