## [Editor Report]

In the canonical model of G protein-coupled receptor desensitization and endocytosis, the unmasking of the β-arrestin C-terminus plays a crucial role. Here, Barsi-Rhyne and colleagues revise and extend this fundamental paradigm by showing that there is a second structural region, the β-arrestin C lobe, that governs GPCR endocytic behavior. Importantly, class A GPCRs that transiently interact with arrestins prefer this new mode while class B GPCRs, which form more stable interactions with arrestins, employ both the canonical and the newly identified endocytic mechanism. Intriguingly, the canonical mode enables prolonged signaling while the new mode promotes rapid desensitization.

---

## [Decision Letter]

**Decision letter after peer review:**

Thank you for submitting your article "Discrete GPCR-triggered endocytic modes enable β-arrestins to flexibly regulate cell signaling" for consideration by *eLife*. Your article has been reviewed by 3 peer reviewers, and the evaluation has been overseen by a Reviewing Editor and Suzanne Pfeffer as the Senior Editor. The following individual involved in the review of your submission has agreed to reveal their identity: Evi Kostenis (Reviewer #3).

Essential revisions:

1. It would be important to carry out some experiments to show it is truly CME and not another route.

2. Determine if this region does or does not bind clathrin – "…using the Tail deletion should remove all characterised CCV motifs so if it still binds clathrin/AP2, there is the answer. If it doesn't bind but is CME then it must be via something else."

*Reviewer #1 (Recommendations for the authors):*

As high-quality data deserve high-quality presentation, a number of issues should be resolved in revision.

1. Line 34. B-arrestins were named because they preferred b2-adrenergic receptors over rhodopsin, in contrast to visual arrestins. That's where the letter "b" came from.

2. Lines 213-215. The text does not agree with the numbers presented. Increased potency means action at lower concentrations, so EC50 of -7.3 means greater potency than EC50 of -6.8.

3. M2 muscarinic receptor was shown to internalize via both caveolae and CCPs. The former route is arrestin-independent (J Biol Chem. 1997 Sep 19;272(38):23682-9), even though M2 binds b-arrestins and is desensitized by them. If that is true, the authors should have observed M2R internalization in b-arrestin1/2 KO cells. Did they?

4. M2R is particularly interesting in another way: this receptor contains phosphorylation sites that arrestins recognize in the 3rd cytoplasmic loop, in contrast to b2AR and other receptors carrying these sites in the C-terminus. The authors should note and discuss this.

5. The effects of disabling clathrin and AP binding sites in b-arrestin CT on GPCR internalization were studied in detail (J Biol Chem. 2002 Aug 23;277(34):30760-8). The authors should compare internalization in the presence of their b-arrestin mutant where both these sites were disabled with that described earlier.

6. There was a report that b-arrestin1 (unlike b-arrestin2) has an additional clathrin-binding site (J Biol Chem. 2009 Oct 23;284(43):29860-72). Did the authors consider or test its role in GPCR internalization?

7. In view of the inability of visual arrestin to promote GPCR internalization, the authors might want to discuss the similarities and differences between available structures of visual arrestin complex with rhodopsin and b-arrestin1 complex with several non-visual GPCRs. This might shed some light on the cross-talk between the receptor-binding surface of arrestins and their opposite cytoplasmic side implied by the data.

8. Some editing is needed. In the abstract, "that each mode has" should be "different modes have".

*Reviewer #2 (Recommendations for the authors):*

I suggest strengthening the science in the following ways:

1. Improve referencing and explanations of existing published data. These include but are not limited to Using correct PLoS reference of Β ear structure from McMahon lab, not PDB access PMID: 16903783

This work is based on that from Traub lab published the year before, which is completely omitted – it must be included PMID: 16516836

Include referencing of the body of work from Royle lab, Graham/Kelly lab, and Traub/Owen labs on identifying and characterising a wide range of signals that can bind to clathrin PMID: 27813245, PMID: 14981508, PMID: 21939487 WxxW

Discuss the relationship between Clobe site and Arrestin2L splice loop clathrin binding site – especially Line144 as this splice loop site is important for Arrestin2L mediated internalisation – are splice loop and Clobe redundant assuming they are different as they appear to be.

2. Show that the Clobe using internalisation events are mediated by CME – use inhibitors commonly available (dynasore, pitstop) or dynamin mutation, KD of clathrin. KD of AP2 would also help to show whether AP2 was necessary, which would add further interesting information.

3. Rather than delete the entire Ctail, which is a crude mutation, replace the LIELD and FxxFxxxR sites individually to better define the role of Clobe.

4. Demonstrate what Clobe binds to. As a minimum, biochemical binding to clathrin should be tested as elegantly carried out in Graham/Kelly lab work. Solving a structure if the interaction is with clathrin is not required but would be a nice addition. The Graham/Kelly work shows that very different sequences can bind to the same sites on clathrin terminal domains.

Doing an IP/binding study from cytosol, analysed by mass spec using β-arrestin 2 and Ctail deleted and Clobe mutated β-arrestin would probably be very informative.

5. For the benefit of readers, an alignment of arrestins 1,2,3 (vArr, B1Arr, B2Arr) along with the chimera should be shown in a main figure with all relevant sites clearly highlighted.

Line108 list the contiguous residues involved in binding and suggest how a conservative mutation of N to T could produce such a profound effect – incidentally please confirm the mutant still folded.

Line150 Discuss possible explanation(s) for this data not agreeing with PMID: 16516836 that the Bear FxxFxxxR site is key for concentrating Β-adrenergic receptor in CCPs.

*Reviewer #3 (Recommendations for the authors):*

It is a true pleasure for me to review such elegant work and to suggest some comments to further enhance the paper's impact.

---

## [Author Response]

Essential revisions:1. It would be important to carry out some experiments to show it is truly CME and not another route.

We agree this is important to demonstrate. We have done so by performing clathrin knockdown experiments and have included this data in Extended Data Figure 1i. The alternate endocytic mode indeed is clathrin-dependent, consistent with the imaging data showing receptor accumulation in CCPs.

2. Determine if this region does or does not bind clathrin – "…using the Tail deletion should remove all characterised CCV motifs so if it still binds clathrin/AP2, there is the answer. If it doesn't bind but is CME then it must be via something else."

This is also an excellent point. To address it, we include new data using pulldowns with clathrin terminal domain and the β-appendage of AP2, the previously known binding determinants for the β-arrestin CT. We verify both and show that removing the β-arrestin CT disrupts both. Therefore we are confident that the alternate endocytic mode does not rely on either of these interactions. These new data are in the Extended Data Figure 1j of the revised manuscript.

Reviewer #1 (Recommendations for the authors):As high-quality data deserve high-quality presentation, a number of issues should be resolved in revision.1. Line 34. B-arrestins were named because they preferred b2-adrenergic receptors over rhodopsin, in contrast to visual arrestins. That's where the letter "b" came from.

This is a useful clarification. We have altered the text to more fully note the historical basis for β-arrestin nomenclature. Line 34 in the revised manuscript.

2. Lines 213-215. The text does not agree with the numbers presented. Increased potency means action at lower concentrations, so EC50 of -7.3 means greater potency than EC50 of -6.8.

We thank the reviewer for pointing out this error. The inadvertent swap in the listing of EC50 values in the text has been corrected. Line 232-235 in the revised manuscript.

3. M2 muscarinic receptor was shown to internalize via both caveolae and CCPs. The former route is arrestin-independent (J Biol Chem. 1997 Sep 19;272(38):23682-9), even though M2 binds b-arrestins and is desensitized by them. If that is true, the authors should have observed M2R internalization in b-arrestin1/2 KO cells. Did they?

This is a good point. In our hands, all of the GPCRs included in the present study internalize in a β-arrestin-dependent manner. These data are in Figure 4 Extended Data a – f. Specifically, for the M2R, the data are in Extended Data Figure 4c.

4. M2R is particularly interesting in another way: this receptor contains phosphorylation sites that arrestins recognize in the 3rd cytoplasmic loop, in contrast to b2AR and other receptors carrying these sites in the C-terminus. The authors should note and discuss this.

We agree that this is an interesting difference. In our hands M2R clusters with the other class B GPCRs tested, including V2R where the relevant phosphorylation is clearly in the receptor tail. We used the tail / core distinction for simplicity and because it has been widely discussed in the arrestin literature, but agree that this dichotomy is too limited to account for M2R. We have added a caveat that the “tail” distinction that we propose may also extend to the 3rd loop in other GPCRs, and that further studies would be needed to test this. line 214-216 of the revised manuscript.

5. The effects of disabling clathrin and AP binding sites in b-arrestin CT on GPCR internalization were studied in detail (J Biol Chem. 2002 Aug 23;277(34):30760-8). The authors should compare internalization in the presence of their b-arrestin mutant where both these sites were disabled with that described earlier.

We agree that there is a lot known about β-arrestin’s endocytic activity from dominant negatives. In our hands observing DN effects require quite high levels of mutant b-arrestin expression, including for the CT mutations described previously. Nevertheless, we have tested our endocytosis-dead CLB + CT mutant and it is indeed inhibitory when overexpressed at high levels in wild type HEK293 cells. We show this data in Extended Data Figure 3b.

We also can confirm the DN effect of b-arrestin CT described earlier, and we think the CLB+CT mutant is stronger, but we would prefer not to make any claims about this because we have not properly adjusted identical expression levels and we don’t know what purpose this would serve. Our main claim is that the CT mutation is able to rescue B2AR internalization in a β-arrestin KO background while the CLB + CT is not, and we believe that this is clearly supported by the data.

6. There was a report that b-arrestin1 (unlike b-arrestin2) has an additional clathrin-binding site (J Biol Chem. 2009 Oct 23;284(43):29860-72). Did the authors consider or test its role in GPCR internalization?

This is an interesting question. By analyzing deletion mutants for genetic rescue, we found that the splice loop (βarr1-ΔSL) is neither necessary nor sufficient for β2V2R internalization. We have added this new data to Extended Data Figure 3d.

We further note that the function of this splice loop in b-arrestin-1 is currently unclear. The article cited by the reviewer clearly shows that this loop can bind the terminal domain of clathrin heavy chain and that it impacts clustering and removal of receptors from the plasma membrane. However, a recently reported structure of β-arrestin-1 bound to the M2R reported that this loop is inserted into the membrane bilayer and also demonstrated that mutation, by replacement of three residues with aspartic acid, impaired recruitment of b-arrestin-1 to plasma membrane and internalization of M2R (PMID 31945772).

7. In view of the inability of visual arrestin to promote GPCR internalization, the authors might want to discuss the similarities and differences between available structures of visual arrestin complex with rhodopsin and b-arrestin1 complex with several non-visual GPCRs. This might shed some light on the cross-talk between the receptor-binding surface of arrestins and their opposite cytoplasmic side implied by the data.

We are aware of 4 available structures of GPCRs bound to either visual or β-arrestins. There appear to be minimal conformational differences in the C-lobe base region of visual and β-arrestin in these structures. We note that all of the available structures are of moderate to low resolution, which is sufficient to define the overall topology of the interaction between arrestin and a GPCR but does not enable us to finely dissect small changes in conformation between visual arrestin and β-arrestins. Even among multiple GPCRs bound to β-arrestin, there is a relatively large degree of conformational heterogeneity when comparing the orientation of arrestin vs. the GPCR 7TM domain. With these caveats, we are not comfortable making structural inferences but agree thus is an interesting question for future investigation.

8. Some editing is needed. In the abstract, "that each mode has" should be "different modes have".

We thank the reviewer for pointing this out and have corrected this error.

Reviewer #2 (Recommendations for the authors):I suggest strengthening the science in the following ways:1. Improve referencing and explanations of existing published data. These include but are not limited to Using correct PLoS reference of Β ear structure from McMahon lab, not PDB access PMID: 16903783This work is based on that from Traub lab published the year before, which is completely omitted – it must be included PMID: 16516836

We thank the reviewer for pointing out this error. We have corrected these citations as indicated.

Include referencing of the body of work from Royle lab, Graham/Kelly lab, and Traub/Owen labs on identifying and characterising a wide range of signals that can bind to clathrin PMID: 27813245, PMID: 14981508, PMID: 21939487 WxxW

We have included a more thorough discussion of the many interactions with clathrin to the revised discussion. Line 304-306 in the revised manuscript.

Discuss the relationship between Clobe site and Arrestin2L splice loop clathrin binding site – especially Line144 as this splice loop site is important for Arrestin2L mediated internalisation – are splice loop and Clobe redundant assuming they are different as they appear to be.

We thank the reviewer for this feedback. We have included an experiment in the supplement that addresses the role of arrestin2L splice loop that binds clathrin and added a sentence describing this result. This new information is in Extended data Figure 3d and Line 156-158. In our hands the splice loop is neither necessary nor sufficient for internalization of the β2V2R.

2. Show that the Clobe using internalisation events are mediated by CME – use inhibitors commonly available (dynasore, pitstop) or dynamin mutation, KD of clathrin. KD of AP2 would also help to show whether AP2 was necessary, which would add further interesting information.

We agree that this is an important point. Knockdown of clathrin heavy chain strongly inhibits β2AR internalization promoted by both wild type βarr2 and truncation mutants lacking the previously identified AP2 and CHC binding sites. These new data are added in Extended data Figure 1i and complement the demonstration of accumulation in CCPs as was already shown.

3. Rather than delete the entire Ctail, which is a crude mutation, replace the LIELD and FxxFxxxR sites individually to better define the role of Clobe.

We agree that it is conceivable that deleting the CT is a bit crude. We also show that Ala substitutions also do not block the endocytic activity (Figure 1), and that mutating the CLB in the context of a wild type CT does block the activity for β2AR (Figure 3). To further address this concern, we have performed additional experiments testing rescue of β2AR internalization after individual mutation of AP2 or clathrin binding motifs. The results, shown below, are fully consistent with the interpretation reached from the results included based on deletion and combined point mutations.

4. Demonstrate what Clobe binds to. As a minimum, biochemical binding to clathrin should be tested as elegantly carried out in Graham/Kelly lab work. Solving a structure if the interaction is with clathrin is not required but would be a nice addition. The Graham/Kelly work shows that very different sequences can bind to the same sites on clathrin terminal domains.Doing an IP/binding study from cytosol, analysed by mass spec using β-arrestin 2 and Ctail deleted and Clobe mutated β-arrestin would probably be very informative.

The reviewer is correct that we presently do not know the binding partner(s) responsible for the CLB activity. We believe identifying this partner is outside of the present scope, but is an interesting and active area of ongoing work.

To address the reviewer’s concern, we have included in the revised manuscript additional data indicating that removal of the βarr2 CT ablates interaction with the known binding determinants in the β-appendage of AP2 and the clathrin terminal domain (Ext Data Figure 1j and line 94-95). Thus we are confident that, whatever the additional interaction(s) are, they are distinct from the previously known interactions which we agree have been elegantly followed up in multiple studies.

5. For the benefit of readers, an alignment of arrestins 1,2,3 (vArr, B1Arr, B2Arr) along with the chimera should be shown in a main figure with all relevant sites clearly highlighted.

We include an alignment of vArr, βarr2 and the chimera in Supp 2. The βarr1 sequence is similar in the relevant regions. We included both β-arrestins in the sequence logo analysis (Supp 3) and note this in the legend to Supp 3 as well as in the Methods. To specifically address the reviewer’s concern, we also add to the revised manuscript an additional sequence alignment focused on the critical region of divergence, including both β-arrestin sequences. As the main figures are already quite dense, and already illustrate the key changes in the context of protein structure, we would prefer to add this information in Ext data Figure 2b.

Line108 list the contiguous residues involved in binding and suggest how a conservative mutation of N to T could produce such a profound effect – incidentally please confirm the mutant still folded.

The possibility of spurious results due to misfolding is a reasonable concern. We are confident that the CLB mutant is indeed folded because it is recruited to the plasma membrane in response to receptor activation and it promotes regulated endocytosis of a subset of GPCRs in genetic rescue experiments. To further verify this, we have carried an additional in vitro study using an scFv that binds to a 3-dimensional epitope which is specific to the activated β-arrestin conformation. We show that the CLB mutant is competent to bind the ScFv in a V2RPP-dependent manner. We include these data also for the ∆CT and CLB,∆CT double mutants. Concentration response analysis reveals increased potency and reduced efficacy of V2RPP, as expected due to the autoinhibitory function of the CT. These new data are in Ext Data Figure 3a. Therefore we are confident that the mutant constructs studied are indeed properly folded.

We agree with the reviewer that it is notable that a conservative substitution has such a profound effect on arrestin function. We previously proposed that this mutation may disrupt a protein-protein interaction leading to the observed functional effect. Still, we agree that the large effect produced by a single, conservative substitution is remarkable. We speculate that this could occur entropically if the small change produces a steric clash, but this is only a speculation and answering this interesting question will require first identifying the partner. In the revised manuscript we explicitly note and discuss this (line 159-162).

Line150 Discuss possible explanation(s) for this data not agreeing with PMID: 16516836 that the Bear FxxFxxxR site is key for concentrating Β-adrenergic receptor in CCPs.

We looked up the reference cited and were unable to find any data regarding clustering of adrenergic receptors. However, we note that another study (PMID: 10770944, already cited) does make this claim explicitly. Our results fully support the ability of the β-arrestin CT to support clustering, and we verify this in Figure 3. We also show that the CT is critical for endocytosis of a subset of GPCRs. Our point is that the CT is not essential for this, at least for some GPCRs, why we conclude there is another mode. To further support this we have added new data in the revised manuscript demonstrating that AP2 pulldown of β-arrestin is abrogated by mutation of the CT. This supports our conclusion that the additional endocytic mode revealed by our rescue strategy utilizes protein interaction interface(s) that are distinct from those mediating the effects of the CT.

We do not think that our results are in conflict with the previous conclusions. We speculate that our ability to detect the distinct importance of the β-arrestin CLB is based on the assays used. No previous studies have assessed the endocytic activity of β-arrestin by rescue from a null background. We do not contest the CT interactions elegantly demonstrated in vitro, or the importance of the CT for clustering and endocytosis of some GPCRs. Rather we report something new that wasn’t detected using these earlier assays. However, we do note that there is previous evidence that β-arrestin can indeed drive internalization of the β2AR without the CT. Evidence supporting this is in a foundational paper (Orsini et al. as cited) and also as discussed above in response to reviewer #1 (critique 5).